# Deoxynivalenol Induces Intestinal Damage and Inflammatory Response through the Nuclear Factor-κB Signaling Pathway in Piglets

**DOI:** 10.3390/toxins11110663

**Published:** 2019-11-14

**Authors:** Xi-Chun Wang, Ya-Fei Zhang, Li Cao, Lei Zhu, Ying-Ying Huang, Xiao-Fang Chen, Xiao-Yan Chu, Dian-Feng Zhu, Sajid Ur Rahman, Shi-Bin Feng, Yu Li, Jin-Jie Wu

**Affiliations:** College of Animal Science and Technology, Anhui Agricultural University, 130 West Changjiang Road, Hefei 230036, China; wangxichun@ahau.edu.cn (X.-C.W.); djdshhz@163.com (Y.-F.Z.); caoli@ahau.edu.cn (L.C.); zhuleizl@ahau.edu.cn (L.Z.); huangyingying@ahau.edu.cn (Y.-Y.H.); chenxiaofang0912@163.com (X.-F.C.); 13856982950@163.com (X.-Y.C.); zdf3793@163.com (D.-F.Z.); dr_sajid226@yahoo.com (S.U.R.); fengshibin@ahau.edu.cn (S.-B.F.)

**Keywords:** deoxynivalenol, intestinal mucosal injury, inflammatory cytokine, nuclear factor-κB signaling, piglets

## Abstract

Deoxynivalenol (DON) is highly toxic to animals and humans, but pigs are most sensitive to it. The porcine mucosal injury related mechanism of DON is not yet fully clarified. Here, we investigated DON-induced injury in the intestinal tissues of piglet. Thirty weanling piglets [(Duroc × Landrace) × Yorkshire] were randomly divided into three groups according to single factor experimental design (10 piglets each group). Piglets were fed a basal diet in the control group, while low and high dose groups were fed a DON diet (1300 and 2200 μg/kg, respectively) for 60 days. Scanning electron microscopy results indicated that the ultrastructure of intestinal epithelial cells in the DON-treated group was damaged. The distribution and optical density (OD) values of zonula occludens 1 (ZO-1) protein in the intestinal tissues of DON-treated groups were decreased. At higher DON dosage, interleukin *(IL)-1β*, *IL-6*, and tumor necrosis factor-α mRNA levels were elevated in the intestinal tissues. The mRNA and protein levels of *NF-κB p65*, *IκB-α*, *IKKα/β*, *iNOS*, and *COX-2* in the small intestinal mucosa were abnormally altered with an increase in DON concentration. These results indicate that DON can persuade intestinal damage and inflammatory responses in piglets via the nuclear factor-κB signaling pathway.

## 1. Introduction

Mycotoxins are secondary metabolites of molds, which are highly toxic to humans and animals. They are frequently found in food and feed that affects biosafety [1]. Deoxynivalenol (DON) is a trichothecene produced by *Fusarium graminearum* and *Fusarium oxysporum*, and exerts various toxic effects, such as neurotoxic and cytotoxic effects. [2,3,4]. DON has a high detection rate worldwide. Mostly feed composition and complete feed are contaminated by DON. DON has been widely detected in feed in various provinces of China. From a total of 573 samples analyzed for DON from 2013 to 2015, 93.9% of the samples contained this mycotoxin at a dose ranging from 0 to 4402.7 μg/kg [5]. It has been shown that DON production is greatly influenced by weather [6,7,8]. In China and Europe, the DON safety standard concentrations for swine feed are 1000 and 900 μg/kg, respectively. The detected value in the survey was much higher than the standard limit. Hence, studies on DON concentrations included in feed are extremely significant. 

There are harmful effects of DON on animal tissues and cells. Studies have shown that DON exerts neurotoxic effects, such as lipid peroxidation and neuronal apoptosis in animal brain tissues [9,10]. Moreover, DON has been proved to significantly affect intestinal epithelial cell 6 (IEC-6) cell viability in the intestinal tract of mice [11]. Pig is the most DON-sensitive species among farm animals [12]. Resistance to toxic substances in the external environment depends primarily on barrier integrity. According to different cells and animal studies, exposure to trichothecene affects intestinal barrier integrity, which leads to increased translocation of harmful stressors [13]. By altering the expression of tight junction (TJ) proteins, intestinal barrier function can be regulated directly or indirectly [14]. Exposure of pigs to 2850 μg/kg of DON for five weeks results in decreased claudin-4 protein expression in the jejunum [15]. Connexins are known to be associated with TJs, and decreased expression of this protein indicates that the intestinal barrier has been damaged. 

In addition to affecting intestinal mucosal integrity, DON can also induce abnormal expression of pro-inflammatory cytokines [16]. A previous study showed that 1 mg/mL of DON results in a significant increase in interleukin (IL)-12, IL-1β, IL-8, IL-6, and tumor necrosis factor-α (TNF-α) level in the intestinal porcine epithelial cell line, IPEC-J2 [17]. DON has also been shown to upregulate TNF-α expression in mammalian cell lines, such as intestinal epithelial cells, as well as the expression of IL-1β and IL-18 in the serum of patients with gastroenteritis. Previous researches indicate that DON can directly induce inflammatory factor secretion in the intestinal epithelium, thereby inducing intestinal inflammation. Furthermore, this effect is thought to be potentially mediated by the nuclear factor-κB (NF-κB) inflammatory pathway, which has a deleterious effect on the host when inappropriately activated. A recent study showed that DON could activate the NF-κB pathway, leading to inflammasome activation [18].

In this study, the minimum 1300 μg/kg amount of DON was used, which is close to the global standard limit. Until now, the mechanism of DON-induced intestinal damage and inflammatory response in piglets has not been elucidated. Here, our findings provide a theoretical basis for the intestinal inflammation and damage induced by a DON diet in piglets.

## 2. Results

### 2.1. Morphological Changes in Intestinal Porcine Epithelial Cells

The morphology of intestinal epithelial cells was observed by transmission electron microscopy, as presented in Figure 1. In the control group, intestinal epithelial cells showed a clear and complete nucleus, with subcellular organelles (Figure 1, C1). The mitochondrial structure is normal, and threadlike cristae can be observed. (Figure 1, C2). Treatment with different concentrations of DON induced morphological changes significantly. In the low dose group (1300 μg/kg DON), severe endoplasmic reticulum swelling and irregular chromatin distribution (Figure 1, L1), along with mitochondrial vacuolization were observed in the cells (Figure 1, L2). In the high dose group (2200 μg/kg DON), chromatin agglutination and karyopyknosis, along with mitochondrial swelling and vacuolization, were clearly detected in the cells (Figure 1, H1). A lesser number of organelles were observed in this group (Figure 1, H2).

### 2.2. Effects of DON on ZO-1 Protein Expression Levels in Intestinal Tissues

The expression of ZO-1 in the intestinal mucosa of each group was decreased with an increasing concentration of DON dosage. The distribution and optical density (OD) values of ZO-1 in the duodenum and jejunum of the DON-treated groups were significantly decreased (*p* < 0.01) compared to those in the control group (Figure 2, Table 1). In the ileum, the distribution and OD values of ZO-1 were significantly lower in the high dose group (*p* < 0.01) compared with the control group (Figure 2, Table 1), and were lower in the low dose group than in the control group (*p* < 0.05).

### 2.3. Effects of DON on the mRNA Expression of Inflammatory Cytokines in Intestinal Tissues

As shown in Figure 3, in the duodenum and ileum, the relative mRNA expression of *IL-1β* in the DON-treated groups increased significantly with an increasing of DON dosage (*p* < 0.01, Figure 3A), while in the jejunum, *IL-1β* mRNA level was significantly higher in the high dose group compared to that with the other two groups (*p* < 0.01, Figure 3A).

In duodenum and jejunum, the mRNA expression of *IL-6* was significantly increased in the high dose group compared to that in the control group (*p* < 0.01, Figure 3B); its expression was more significantly increased in the high dose group than in the low dose group (*p* < 0.01 and *p* < 0.05, Figure 3 B). Further, *IL-6* mRNA expression was significantly increased in the ileum of the DON-treated groups compared to that in the control group (*p* < 0.01, Figure 3B).

The relative mRNA expression of *TNF-α* in a different intestinal segment of the DON-treated groups increased with an increasing dosage of DON (*p* < 0.05, Figure 3C). In the duodenum, *TNF-α* expression was significantly increased in the high dose group compared to that in the control group (*p* < 0.01, Figure 3C), and in the jejunum and ileum, its expression was significantly higher in the high dose group compared to the other two groups (*p* < 0.01, Figure 3C). 

The relative mRNA expression of *IL-1β*, *IL-6*, and *TNF-α* was increased with an increase of DON dosage, indicating that DON can induce inflammatory responses in the small intestinal mucosa.

### 2.4. Effects of DON on the Protein Expression of NF-κB Signaling Pathway-Related Molecules

The protein expression levels of NF-κB pathway-related molecules in the small intestine of piglets were determined, as shown in Figure 4. The protein bands in the different intestinal segments of piglets are shown in Figure 4A. In the duodenum and ileum, NF-κB p65 protein expression in the DON-treated groups increased significantly with an increase of DON dosage (*p* < 0.01, Figure 4B), while, in the jejunum, its expression was significantly higher in DON-treated groups than that in the control group (*p* < 0.01, Figure 4B).

The phosphorylation levels of NF-κB p65 in the intestine of DON-treated groups were significantly increased with an increasing dosage of DON (*p* < 0.05 or *p* < 0.01, Figure 4C), and were significantly increased in both DON-treated groups compared to that in the control group (*p* < 0.01, Figure 4C).

The ratio of IκB-α phosphorylation to non-phosphorylation is shown in Figure 4D. In the duodenum and ileum, the ratio was significantly increased with a high dosage of DON (*p* < 0.01, Figure 4D), while, in the jejunum, the ratio was significantly increased in the high dose group compared to those in the other two groups (*p* < 0.01, Figure 4D).

In the duodenum and ileum, the protein expression of cyclooxygenase 2 (COX-2) was significantly increased with an increase of DON dosage (*p* < 0.01, Figure 4E), while, in the jejunum, its expression was significantly increased in both DON-treated groups compared to that in the control group (*p* < 0.01, Figure 4E).

### 2.5. Effects of DON on the mRNA Relative Expression of NF-κB Signaling Pathway-Related Molecules

As shown in Figure 5, the relative mRNA expression of *NF-κB p65* significantly increased with an increase of DON dosage in the duodenum and ileum (*p* < 0.01, Figure 5A), while, in the jejunum, its expression in the high dose group was higher than that in the low dose group (*p* < 0.05, Figure 5A), and significantly higher than that in the control group (*p* < 0.01, Figure 5A).

The relative mRNA expression of *IκB-α* in the duodenum of the high dose group was significantly decreased compared to those in the other two groups (*p* < 0.01, Figure 5B), and was significantly lower in the low dose group than in the control group (*p* < 0.05, Figure 5B). *IκB-α* mRNA expression in the jejunum and ileum of both DON-treated groups was significantly decreased compared to that in the control group (*p* < 0.01, Figure 5B). *IκB-α* mRNA expression in the jejunum of the high dose group was significantly decreased compared with that in the low dose group (*p* < 0.05, Figure 5B).

In the duodenum, the relative mRNA expression of *IKKα* significantly increased with an increase of DON dosage (*p* < 0.01, Figure 5C). In the jejunum and ileum, its expression was increased in the high dose group compared to that in the low dose group (*p* < 0.05 or *p* < 0.01, Figure 5C), and was significantly increased in the high dose group compared to that in the control group (*p* < 0.01, Figure 5C).

The relative mRNA expression of *IKKβ* was increased in the duodenum and ileum of the high dose group compared to those in the other two groups (*p* < 0.01, Figure 5D). In the duodenum, its expression was higher in the low dose group than in the control group (*p* < 0.05, Figure 5D). In the jejunum, its expression was significantly higher in both DON-treated groups than in the control group (*p* < 0.01, Figure 5B). *IKKβ* mRNA expression in the jejunum was increased in the high dose group compared to that in the low dose group (*p* < 0.05, Figure 5D). 

As shown in Figure 5E, in the duodenum and jejunum, the relative mRNA expression of *iNOS* in both DON-treated groups increased significantly with an increase of DON dosage (*p* < 0.01, Figure 5E). *iNOS* mRNA expression was increased in the ileum of the high dose group compared to that in the control group (*p* < 0.05, Figure 2).

The relative mRNA expression of *COX-2* in the duodenum and jejunum of DON-treated groups increased with an increase of DON dosage (*p* < 0.01, Figure 5F). *COX-2* mRNA expression was significantly increased in the ileum of both DON-treated groups compared to that in the control group (*p* < 0.05 or *p* < 0.01, Figure 5F), and was increased in the high dose group compared to that in the low dose group (*p* < 0.05, Figure 5F).

## 3. Discussion

DON, produced by molds, can widely contaminate various crops such as wheat and corn. Previously it was reported that DON could cause food refusal and organ damage, as well as increase the risk of several diseases [19,20,21]. Gastrointestinal injury has been shown to result in intestinal mucosal damage. Microscopic observation of the small intestinal mucosa in immature sows revealed that a low dose of DON (12 μg/kg) induced the number of intestinal villus epithelial cells and lamina propria lymphocytes [22]. This study shows that low and high dosage of DON induced submicroscopic structural damage of intestinal epithelial cells, condensation of chromatin, mitochondrial vacuolization, reduction of granule particles, and widening of cell spaces, indicating that the cytotoxic effect of DON is concentration-dependent.

An extended cell gap suggests that the tight junctions (TJs) between the cells may be broken. TJ is the main structure that restricts the passage of foreign substances across the intestinal mucosa, including occludens and claudins [23]. A research found that DON changed the expression of claudins and reduced its barrier function in IPEC-1 or human intestinal Caco-2 cell line [15]. ZO-1, a member of the membrane-associated guanylate kinase protein family, serves as a scaffold to organize transmembrane TJ proteins and plays a key role in retaining intestinal barrier integrity and TJs during pathological injury [23]. Ingestion of food or feed with high concentrations of DON may cause intestinal damage and affect human and animal health. As a crucial barrier for exogenous toxins from entering the body, the intestines process the mycotoxin from the luminal side of mucosa-protected, as well as absorbed, toxins that reach the cells from the basolateral side via the bloodstream [14]. In this study, the expression of ZO-1 protein in the intestinal mucosa of piglets that were exposed to low and high doses of DON decreased significantly, indicating that the DON concentrations used in this study could induce intestinal mucosal damage and increase intestinal permeability.

In addition to barrier integrity, the innate immune function of intestinal mucosa can also exert immune regulatory effects by producing mucus-containing immunoglobulins and regulating the secretion of inflammatory molecules to protect the body [24]. Previous reports have shown that DON can induce intestinal immune regulation and increase the sensitivity of intestinal cells to inflammation [25,26]. In addition, pro-inflammatory cytokines, including TNF-α and IL-1β, have been shown to increase the permeability of intestinal TJ and promote the inflammatory process by allowing increased antigen penetration. IL-1 is a potent pro-inflammatory cytokine that leads to many inflammatory diseases by stimulating immune and inflammatory responses [27]. Among them, IL-1β plays an essential role in the stimulation and amplification of inflammatory response [28]. IL-6 exerts many pro-inflammatory effects and has been shown to play an important role in the process of intestinal inflammation [29]. TNF-α, a key inflammatory cytokine, is a cell-signaling protein involved in systemic inflammation [30]. In this study, the relative mRNA expression of *IL-1β*, *IL-6*, and *TNF-α* significantly increased in the intestinal tissues of DON-treated groups, indicating that DON induces an inflammatory response in the small intestinal mucosa [31,32]. Hence, our results suggest that the DON concentrations used in this study can induce inflammation in the small intestinal mucosa of piglets, thereby affecting growth.

Studies have shown that long-term or acute exposure to DON induces inflammatory changes in human intestinal epithelial cells (Caco-2) in vitro [33]. Among the inflammatory pathways, the transcription factors of the NF-κB family play a vital role in inflammation and innate immunity [34,35]. When NF-κB is not activated, it is mainly stored in the cytoplasm in the form of p50/65-IκBα dimer [36]. When stimulants, such as pro-inflammatory factors, activate the NF-κB pathway, IKK-dependent phosphorylation and degradation of IκB inhibitor protein is induced. In Caco-2 cells, TNF-α has been shown to increase TJ permeability through activation of NF-κB signaling, resulting in the downregulation of ZO-1 protein expression, and thereby inducing intestinal epithelial barrier dysfunction [37]. DON has also been reported to stimulate mitogen-activated protein kinases (MAPKs) in intestinal epithelial cells and the secretion of its downstream inflammatory mediators. It has been found that NF-κB activity increases in a dose-dependent manner, and increases the phosphorylation of IκB, thus causing or exacerbating enteritis [38]. In addition, some studies have found that the transcriptional activation of COX-2 is regulated by the NF-κB pathway [39,40], thereby demonstrating its important role in intestinal inflammatory responses. In addition to COX-2, activation of NF-κB also upregulates the expression of iNOS. However, the production of iNOS and NO plays an inhibitory role in the activation of NF-κB [41]. In our study, the protein expression of p65, IκB-α, IKKα/β, COX-2, and iNOS and the ratio of phosphorylation and non-phosphorylation of p65 and IκB-α protein were examined in the intestinal mucosa of DON-treated groups. The present results show that DON could activate the NF-κB pathway in the small intestinal mucosal cells, causing abnormal expression of related genes and proteins. A previous study [42] revealed that DON-induced intestinal inflammation in humans is mediated through the activation of the PKR/Hck/MAPK/NF-κB pathway, which is in accordance with the results of our study. 

In summary, the DON concentrations used in this study were found to induce intestinal mucosal damage in piglets. The higher the DON concentration, the greater was the damage in the submicroscopic structure of intestinal epithelial cells. Moreover, DON treatment aggravated inflammatory responses by upregulating the expression of intestinal inflammatory factors via activating the NF-κB signaling pathway. 

## 4. Materials and Methods 

### 4.1. Chemical and Reagents

All the chemicals used in this study were of the highest grade and purity. DON standard was purchased from Sigma Chemical Co. (CAS No. D0156-5MG; St. Louis, MO, USA). TRIzol reagent was obtained from Invitrogen Biotechnology Co., Ltd. (Shanghai, China). The SuperScript III kit and SYBR qPCR mix were obtained from Thermo Fisher Scientific (Waltham, MA, USA). The primary antibody for ZO-1 was purchased from Abcam (Cambridge, MA, USA). The protein assay kit was purchased from Beyotime Institute of Biotechnology (Nanjing, China). The polyclonal antibody for NF-κB p65 was purchased from Proteintech (Rosemont, Chicago, IL, USA). The monoclonal antibody for IκB-α was obtained from Bio-Techne China Co., Ltd. (Minneapolis, MN, USA). Monoclonal antibodies against p-NF-κB p65, p-IκB-α, and COX-2 were purchased from Santa Cruz Biotechnology (Santa Cruz, CA, USA). The β-actin antibody was obtained from Beijing Ray Antibody Biotech (Beijing, China). Diaminobenzidine (DAB) was obtained from Beijing Zhongshan Jinqiao Biotechnology (Beijing, China).

### 4.2. Method Used for Extraction DON

The DON used for the DON feed was cultured and extracted (refer to Jiang’s report [43])*,* and *Fusarium graminearum* spore liquid was inoculated in maize medium to produce toxin at 27 ℃ and a relative humidity of 30% for 25 days. Toxigenic media were collected for preparing the crude toxins extracting solution, eluted twice through chromatography columns, and elution liquid was collected segmented. DON was determined by thin layer chromatography (TLC), and the product was purified by using the multiple crystallization method and confirmed by high performance liquid chromatography (HPLC).

### 4.3. Preparation of DON Feed 

The basal feed was mixed with 1300 μg/kg and 2200 μg/kg of DON according to the stepwise magnification method for the preparation of the DON feed. The chromatographic curves of DON feed purified by HPLC were the same as those of DON standard, indicating that DON and other toxins were not present in the basal feed. 

### 4.4. Animal Care and Diet

All animal experiments and procedures were approved by the Anhui Agricultural University Animal Care Committee (ZXD-L2017425). Thirty cross-bred weanling piglets [(Duroc × Landrace) ×Yorkshire] with a mean body weight of 6.87 ± 0.41 kg were selected from three litters in the Wuxing Agricultural Group of Anhui Province, and randomly divided into 3 groups (10 piglets in each group): control group (basal diet), low dose group (1300 μg/kg DON diet), and high dose group (2200 μg/kg DON diet). The experimental trials lasted for 60 days. The piglets were housed in a mechanically ventilated nursery room with a 12:12 h light/dark cycle. The temperature was maintained at 26–28 °C, with a relative humidity of 50%–60%. The diet provided during the experiment (Table 2) was formulated to meet or exceed the nutritional requirements determined by the National Research Council (NRC, 1998) [44].

### 4.5. Experimental Design and Sample Collection

After 60 days of dietary exposure to DON, five piglets from each group were selected and anesthetized by intramuscular injection of an anesthetic mixture (haloperidol, dihydroetorphine, and 2, 4-dimethylaniline thiazole) at a dose of 0.8 mL/kg. The piglets were sacrificed for the collection of samples. Part of the duodenum was fixed in 3% glutaraldehyde for 12 h, and intestinal injury was detected by electron microscopy. The remaining parts of the duodenum, jejunum, and ileum were divided into two portions. One portion was fixed in 4% paraformaldehyde solution to measure intestinal ZO-1 protein expression by immunohistochemistry. The other portion was preserved in liquid nitrogen and used for detection of the mRNA levels of inflammatory cytokines by quantitative real-time polymerase chain reaction (qRT-PCR).

### 4.6. Analysis of Ultrastructure of Intestinal Porcine Epithelial Cells

The intestinal mucosa was trimmed into 1 mm^3^ tissue blocks by two parallel blades and fixed in 2.5% glutaraldehyde at 4 °C for 4 h, followed by dehydration, soakage, embedding, ultra-thin sectioning, lead citrate staining, and washing. The ultrastructure of intestinal epithelial cells was visualized using the TEOL-2010 high resolution transmission electron microscope (Electronics Corporation, Tokyo, Japan).

### 4.7. qRT-PCR

Total RNA was extracted from the liquid nitrogen-pulverized intestinal tissue sample using TRIzol reagent according to the manufacturer’s instructions. The RNA concentration was determined using NanoDrop Lite (Thermo Fisher Scientific, USA). Reverse transcription was performed using SuperScript III First-Strand cDNA Synthesis Mix (Thermo Fisher Scientific, Waltham, MA, USA). qRT-PCR was performed with SYBR Green qPCR Master Mix (Thermo Fisher Scientific, Waltham, MA, USA). Each sample was assayed 3 times. The PCR reaction was performed on a 7900 Fast Real-Time PCR System (Applied Biosystems, Foster City, CA, USA). The PCR conditions were as follows: 1 cycle at 95 °C for 120 s, 40 cycles at 94 °C for 20 s, 60 °C for 20 s, and 72 °C for 30 s. The relative gene expression levels were calculated according to the 2^−ΔΔCT^ method. The mRNA levels were quantified using *β-actin* as a housekeeping gene for normalization. The primer sequences were synthesized by Sangon Biotech (Shanghai, China) and are listed in Table 3.

### 4.8. Detection of Protein Expression by Immunohistochemistry

Sections (4 µm-thick) were cut from the paraffin blocks. The sections were dewaxed in xylene and then rehydrated with graded ethanol. Antigen recovery with 10 mmol/L citric acid buffer (pH 6.0; Beijing Zhongshan Jinqiao Biotechnology) and microwave heating. Incubation with 3% H_2_O_2_ in methanol for 10 min at 37 °C to inhibits endogenous peroxidases. The paraffin-embedded sections were subsequently incubated in the dark with 10% bovine serum albumin (BSA) prior to incubation with primary antibody diluted in phosphate buffered saline (PBS) for 12–16 h at 4 °C. Secondary antibodies were used to detect the bound antibodies (Zhongshan Golden Bridge Biotechnology, Beijing, China). Color development was performed using 0.05% DAB substrate solution (Beijing Zhongshan Jinqiao Biotechnology, Beijing, China). In the negative control group, PBS was used instead of the primary antibody under the same experimental conditions. 

ZO-1 protein expression was analyzed by Image-Pro Plus 6.0 image analysis software (Media Cybernetics, WA, USA) using the images acquired from the immunohistochemistry experiments. Five images from each of the three groups were randomly selected to analyze the average OD value of the relative staining positive rate of ZO-1 protein positive expression. The final data of each group was the average OD value of ZO-1 protein expression in the five images from five piglets (*n* = 5). The optimal tissue slice location was selected based on the integrity of the cells and tissues in the positive expression zone.

### 4.9. Protein Extraction and Western Blot Analysis

A 0.1 g tissue was weighed and transferred to a 10 mL centrifuge tube. RIPA lysis buffer of 1 ml was added to each tube. The intestinal tissues were homogenized using an FSH-2 adjustable high-speed homogenizer (Jiangsujinyi Instrument, Changzhou, China). The above procedures were carried out on ice. Samples were then incubated on ice for 30 min, followed by centrifugation at 12,000 rpm at 4 °C for 5 min. After collection of the supernatant, the protein assay kit was used to determine the protein concentration (Beyotime Institute of Biotechnology, Nanjing, China). Equal amounts of protein (50 μg) were resolved on 10% sodium dodecyl sulfate-polyacrylamide gels and then transferred to a polyvinylidene difluoride membrane (Merck Millipore, Germany). Membranes were blocked in 5% non-fat milk or BSA (Biosharp, Hefei, China) in TBST at 37°C for 4 h. After being washed three times with TBST, membranes were probed overnight at 4 °C with primary antibodies and anti-β-actin antibody (diluted in TBST). After being washed three times with TBST, the second antibody (1:3000) coupled with horseradish peroxidase-conjugated was incubated at 25 °C for 1 h. The blots were visualized using the SuperSignal™ West Femto Substrate trial kit (Thermo Fisher Scientific, Waltham, MA, USA) and imaged using a Bio-Rad ChemiDoc XRS system (Bio-Rad, Hercules, CA, USA).

### 4.10. Statistical Analysis

All data were presented as means ± standard deviation (*n* = 5). Statistical analysis was carried out by using the Statistical Program for Social Sciences (SPSS) software version 19.0 (IBM Corporation, Armonk, NY, USA). ANOVA was used for the comparison of multiples group. Immunohistochemistry images were analyzed by the Image-Pro Plus 6.0 software to determine the average optical density (OD) values of protein expression. The GraphPad Prism version 5.0 software (GraphPad Software Inc., San Diego, CA, USA) was used to obtain histograms. The Student Newman Keuls post-hoc test was used to analyze protein and mRNA expression. A *p*-value < 0.05 was considered statistically significant.

## Figures and Tables

**Figure 1 toxins-11-00663-f001:**
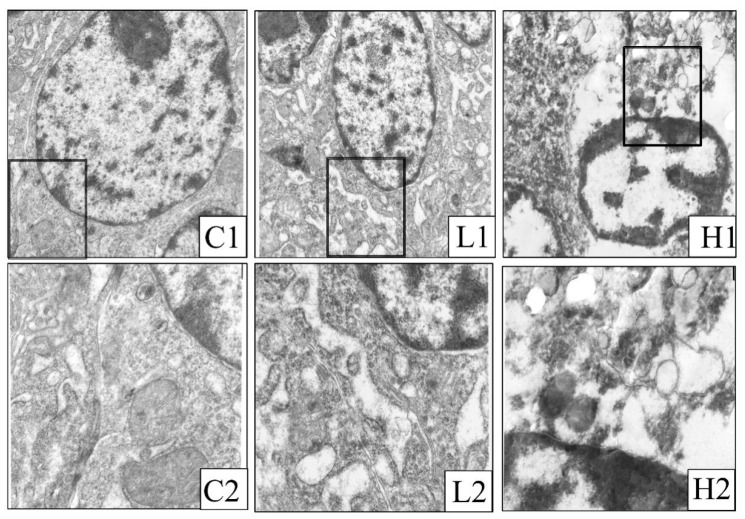
Ultrastructural map of intestinal epithelial cells. (**C1**, **C2**: Control group); (**L1**, **L2**: 1300 μg/kg deoxynivalenol (DON) group); and (**H1**, **H2**: 2200 μg/kg DON group). Image magnification of C1, L1, and H1 is 8000×; image magnification of C2, L2, and H2 is 20,000×. The black boxes in C1, L1, and H1 are enlarged to C2, L2, and H2, respectively.

**Figure 2 toxins-11-00663-f002:**
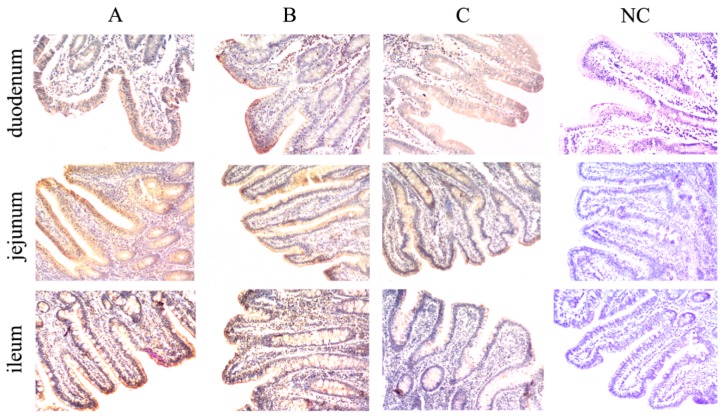
Effect of DON on ZO-1 expression in the intestinal tissues of piglets (the stained sections were photographed at 100× magnification). The letters on the figures indicate: **A**, control group; **B**, low dose group; **C**, high dose group; **NC**, negative control.

**Figure 3 toxins-11-00663-f003:**
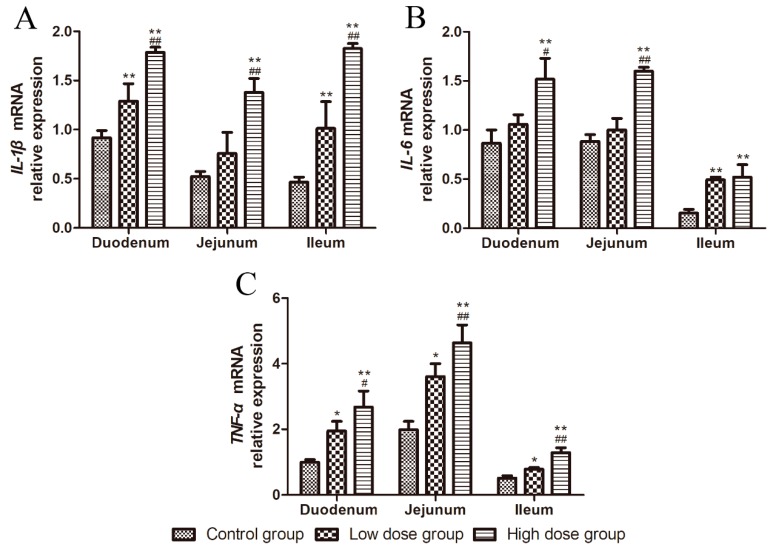
Effects of DON on the relative mRNA expression of inflammatory cytokines in the intestinal tissues. (**A**–**C**) *IL-1β, IL-6,* and *TNF-α* expression in the duodenum, jejunum, and ileum. All data are presented as means ± standard deviation of three independent experiments (*n* = 5). * *p* < 0.05 and ** *p* < 0.01 versus the control group. # *p* < 0.05 and ## *p* < 0.01 versus the low dose group.

**Figure 4 toxins-11-00663-f004:**
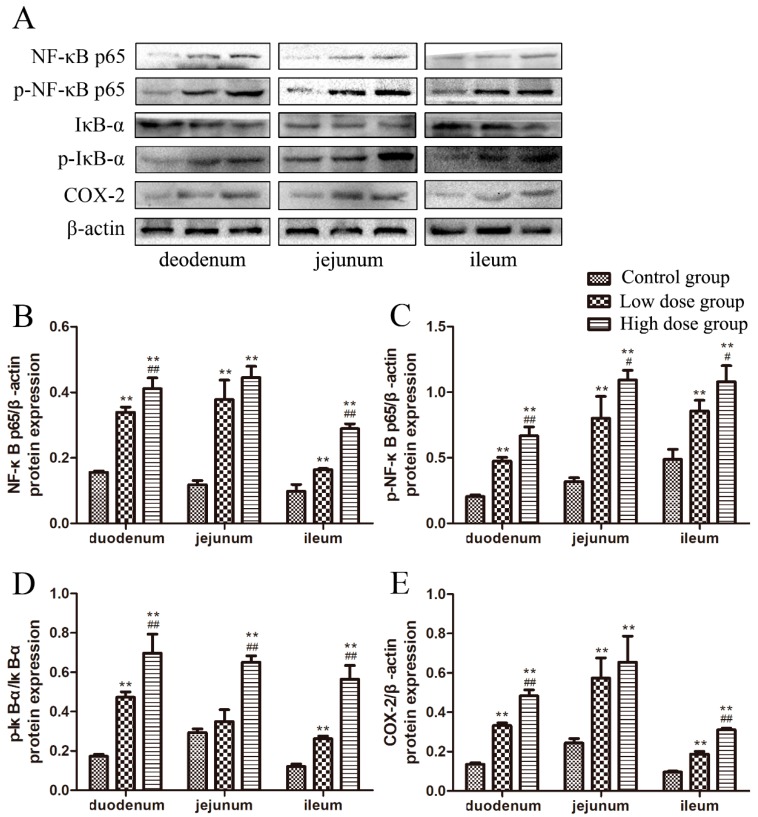
Effects of DON on the relative protein expression level of NF-κB signaling pathway-related molecules. (**A**) Western blotting showing NF-κB p65, p-NF-κB p65, IκB-α, p-IκB-α, COX-2, and β-actin protein levels in the duodenum, jejunum, and ileum. (**B**–**E**) Impact of DON on the protein expression of NF-κB p65, p-NF-κB, p-IκB-α, and COX-2 in the duodenum, jejunum, and ileum. All data are presented as means ± standard deviation of three independent experiments (*n* = 5). ** *p* < 0.01 versus the control group. # *p* < 0.05 and ## *p* < 0.01 versus the low dose group.

**Figure 5 toxins-11-00663-f005:**
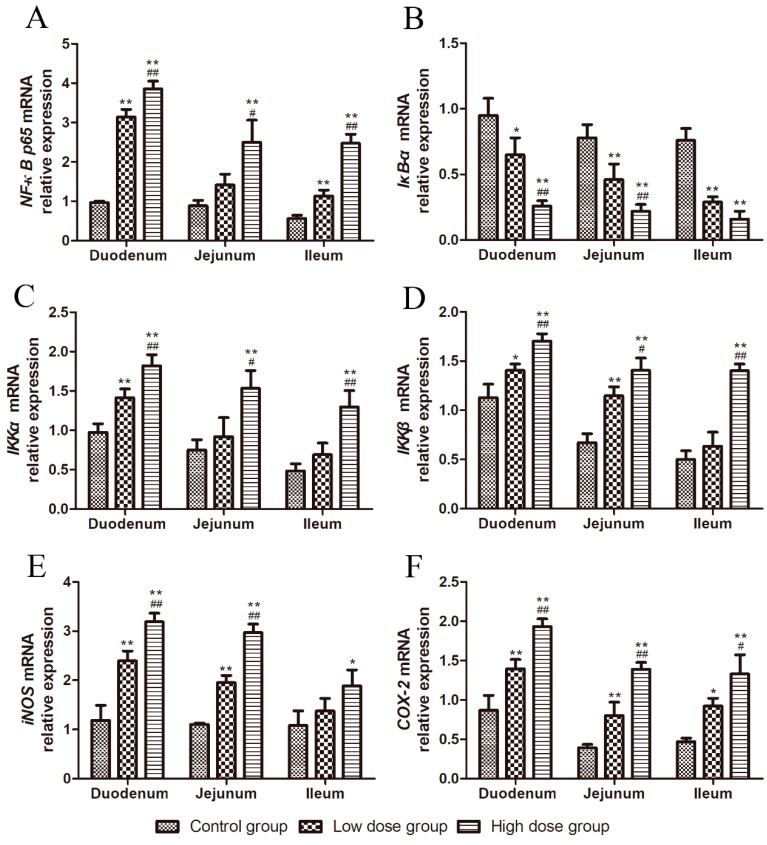
Effects of DON on the relative mRNA expression of NF-κB signaling pathway-related molecules. (**A**–**F**) *NF-κB p65, IKKα, IKKβ, iNOS,* and *COX-2* expression in the duodenum, jejunum, and ileum. All data are presented as means ± standard deviation of three independent experiments (*n* = 5). * *p* < 0.05 and ** *p* < 0.01 versus the control group. # *p* < 0.05 and ## *p* < 0.01 versus the low dose group.

**Table 1 toxins-11-00663-t001:** Average optical density of intestinal ZO-1 protein in piglets.

Index	Control Group	Low Dose Group	High Dose Group
Duodenum	0.1755 ± 0.0515	0.0927 ± 0.0434 **	0.0745 ± 0.0273 **
Jejunum	0.2299 ± 0.0763	0.1377 ± 0.0347 **	0.1287 ± 0.0431 **
Ileum	0.1748 ± 0.0753	0.1184 ± 0.0298 *	0.1110 ± 0.0100 **

All the data are presented as means ± standard deviation of three independent experiments (*n* = 5). * *p* < 0.05 and ** *p* < 0.01 versus the control group.

**Table 2 toxins-11-00663-t002:** Ingredients and nutrient contents of experimental diets.

Experimental Diets	Ingredients (%)
Corn	60.85
Soya bean meal	25.00
Whey powder	5.00
Fish meal	5.00
Calcium hydrogen phosphate	2.20
Limestone	0.69
Bran	0.37
NaCl	0.25
Premix *	0.49
Choline chloride	0.15
Total	100.00
Nutrient levels (%) ^#^	
Crude protein	20.80
Calcium	0.64
Phosphorus	0.51
Lysine	1.06
DE [MJ/Kg] (calculated)	13.50

Notes: * Provided per kg diet: vitamin A, 5250 IU; vitamin D_3_, 1050 IU; vitamin E, 4.5 mg; vitamin K_3_, 1.2 mg; vitamin B_1_, 0.375 mg; vitamin B_2_, 1.8 mg; vitamin B_6_, 0.15 mg; vitamin B_12_, 7.5 µg; niacin, 6 mg; calcium pantothenate, 3.75 mg; folic acid, 0.15 mg; biotin, 7.5 µg; lysine, 0.75 mg; antioxidant, 45 µg; enzyme preparation, 1000 mg; flavor agents, 40 mg; sweet agents, 40 mg; neomycin, 20 mg; Cu, 15 mg; Fe, 144 mg; Zn, 110 mg; Mn, 10.18 mg; I, 0.4 mg; Se, 0.3 mg. ^#^ The digestible energy is calculated while the other nutrients are analyzed.

**Table 3 toxins-11-00663-t003:** Sequences of primers for inflammatory cytokines and the β-actin gene.

Gene	GeneBank Accession No.	Primers	Sequences (5′→3′)	Product Size (bp)
*β-actin*	AY550069.1	Forward	CTGGACTTCGAGCAGGAGATGG	168
		Reverse	TTCGTGGATGCCGCAGGATTC	
*IL-1β*	NM_001302388.2	Forward	TGTGATTGTGGCAAAGGA	111
		Reverse	TCAAGGACGATGGGCTCT	
*IL-6*	NM_214399.1	Forward	GGCAAAAGGGAAAGAATCCAGAC	197
		Reverse	CATCAATCTCAGGTGCCCCA	
*TNF-* *α*	X57321.1	Forward	TGGCCCAAGGACTCAGATCA	107
		Reverse	GGCATACCCACTCTGCCATT	
*NF-κB p65*	KY369935.1	Forward	TCATCGAGCAGCCCAAGCA	240
		Reverse	CAGCCTCATAGAAGCCATCCC	
*IκB-α*	NM_001005150.1	Forward	AGACTCGTTCCTGCACTTGG	201
		Reverse	TCTCGGAGCTCAGGATCACA	
*iNOS*	NM_001143690.1	Forward	GGGTCAGAGCTACCATCCTC	114
		Reverse	CGTCCATGCAGAGAACCTTG	
*IKK* *α*	NM_001114279.1	Forward	CACTCTTACAGCGACAGCAC	145
		Reverse	CCACCTTGGGCAGTAGATCA	
*IKK* *β*	NM_001099935.1	Forward	ACCTGGCTCCCAACGACTT	184
		Reverse	AGATCCCGATGGATGATTCTG	
*COX-2*	MG837549.1	Forward	TGCACGGCGGCAATATTAAA	156
		Reverse	AGTGGAAGTGTGCGACTACA	

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
