# Peer review of "Deoxynivalenol Induces Intestinal Damage and Inflammatory Response through the Nuclear Factor-κB Signaling Pathway in Piglets"

_toxins, 2019, doi:10.3390/toxins11110663_

Round 1

Reviewer 1 Report

As I mentioned previously, this is very important study where the authors investigated the mechanism of DON-induced intestinal damage in piglets and suggested that this mycotoxin may induce inflammation in small intestine tissue by damaging NF-κB signaling pathway. The authors addressed properly my comments, therefore the article could be accepted in the present form.

Minor remark - line 7: "...investigated DON-induced injury in piglet intestinal tissues." Please correct the sentence.

Author Response

Point 1: Minor remark - line 7: "...investigated DON-induced injury in piglet intestinal tissues." Please correct the sentence.

Response 1: Thank you for your valuable suggestions. We have revised the sentence according to the reviewer's suggestion.

Reviewer 2 Report

Dear authors,

I appreciate all the changes in the manuscript that was made, but the main errors and misunderstandings remained unexplained. Reference to the description of results from this experiment published:” Effects of deoxynivalenol on growth performance, serum cytokine and intestinal immune levels in weanling piglets. Journal of Nanjing Agricultural University, 2018, 41(6): 1126-1132).” makes the situation even more complicated and incomprehensible. Unfortunately, the data used in this publication are not consistent with those used in the evaluated manuscript.

The description of materials and methods was changed, “We purchased the DON standard from Sigma Chemical Co. The DON used for the DON feed was cultured and extracted by us, according to the method in the following reference:(Jiang Y J, Xu w, et al. Extraction, purification and concentration determination of deoxynivalenol. Chinese Journal of Veterinary Science, 2017, 37(9):1771-1777.”. That's still not the right way to describe it (please indicate at least an outline of the methodology). The fact that these articles are in Chinese also does not make things easier.

The doses used in the experiment are different in different places (line 270 and line 279) (μg / kg of feed vs. μg / kg b.w). Presented manuscript suggest that experimental pigs eat as much feed as they weigh.

Fig. 2 - line 92, despite the fact that it is still of very poor quality - shows that the experimental group was not exposed to such a high toxic dose (assessment of the histopathological changes).

Line 49 - 'Exposure of pigs to 2850 μg / kg bw DON has been shown to reduce claudin-4 protein expression in the jejunum for five weeks [15]. "- The mentioned article does not contain such data and research was carried out on tissue explants (Pinton et all).

All this suggests that the results presented were developed in a very unprofessional manner and are not suitable for publication.

Author Response

Point 1: Reference to the description of results from this experiment published:” Effects of deoxynivalenol on growth performance, serum cytokine and intestinal immune levels in weanling piglets. Journal of Nanjing Agricultural University, 2018, 41(6): 1126-1132).” makes the situation even more complicated and incomprehensible. Unfortunately, the data used in this publication are not consistent with those used in the evaluated manuscript.

Response 1: Thank you for your valuable comments. The data difference between the published Chinese article and this manuscript is the concentration difference of DON. Among them, DON concentration in the published Chinese article is the calculated value per kilogram of feed, while the DON concentration in this manuscript is the actual measured value per kilogram of feed. In order to make the test data more representative, the actual measured value is selected.

Point 2: The description of materials and methods was changed, “We purchased the DON standard from Sigma Chemical Co. The DON used for the DON feed was cultured and extracted by us, according to the method in the following reference:(Jiang Y J, Xu w, et al. Extraction, purification and concentration determination of deoxynivalenol. Chinese Journal of Veterinary Science, 2017, 37(9):1771-1777.”. That's still not the right way to describe it (please indicate at least an outline of the methodology). The fact that these articles are in Chinese also does not make things easier.

Response 2: Thank you very much for your valuable comments. We have added a detail outline of the methodology of cultured and extracted DON following your valuable suggestion in the revised manuscript. Refer to 4.2 for detail.  

Point 3: The doses used in the experiment are different in different places (line 270 and line 279) (μg / kg of feed vs. μg / kg b.w). Presented manuscript suggest that experimental pigs eat as much feed as they weigh.

Response 3: Thank you for your careful review. The purpose of this study is to evaluate the intestinal injury and inflammatory response of piglets caused by 1300 μg DON and 2200 μg DON per kilogram of feed. The expression has been modified.

Point 4: Fig. 2 - line 92, despite the fact that it is still of very poor quality - shows that the experimental group was not exposed to such a high toxic dose (assessment of the histopathological changes).

Response 4: Thank you for your valuable comments. In this manuscript, it can be seen from Figure 2 that the positive expression area of ZO-1 protein decreased in DON-treated groups compared to those in the control group. The data analysis results are shown in Table 1.

Point 5: Line 49 - 'Exposure of pigs to 2850 μg / kg bw DON has been shown to reduce claudin-4 protein expression in the jejunum for five weeks [15]. "- The mentioned article does not contain such data and research was carried out on tissue explants (Pinton et all).

Response 5: Thank you for your valuable comment and important point. Because I misunderstood the opinion of the reviewer before, I changed the unit from μg / kg to μg / kg b.w directly, which resulted in the inconsistency of DON contents. The original is ‘Exposure of pigs to 2850 μg/kg of DON for five weeks was shown to results in the decreased claudin-4 protein expression in the jejunum’, content in accordance with reference 15.

Reviewer 3 Report

  This paper clarified physiological responses in piglet intestinal tissues which is treated with deoxynivalenol. The results clearly showed that several cytokine mediators and intracellular signaling molecules relating to inflammatory responses were stimulated in them. The experiments were well conducted. The results are reasonable and consistent with analyses using cultured cells previously reported. As the results of this could be predictable in terms of former reports, it cannot be said that it is completely new knowledge. However, this study is of good significance that the domestic animals were practically used. I can substantially recommend this paper to be published in this journal. However, I would like to ask authors' responses and corrections for the following minor points:   Although this kind of research has been reported as precedents, some of them are inconsistent with this report. Pasternak et al. (Toxins, 2018, 10, 40) reported that interleukin (IL) IL1β, IL-8, IL-13, tumor necrosis factor (TNF) -α, or interferon (IFN) -γ showed no significant difference in weaner pigs with or without exposures to DON-contaminated diet. How do the authors think about this?   The sections 2.3, 2.4 and 2.5, and the caption of Figure 4  repeat similar sentences and appear redundant. Could the authors improve these texts a little more?

Author Response

Point 1: Although this kind of research has been reported as precedents, some of them are inconsistent with this report. Pasternak et al. (Toxins, 2018, 10, 40) reported that interleukin (IL) IL1β, IL-8, IL-13, tumor necrosis factor (TNF) -α, or interferon (IFN) -γ showed no significant difference in weaner pigs with or without exposures to DON-contaminated diet. How do the authors think about this?

Response 1: Thank you for your careful review of our manuscript. First of all, the experimental time of the two articles is very different; Pasternak's article shows that the experimental time is 25 days, while the experimental period of this study is 60 days, in which the dose and mode of exposure are also different. In Pasternak's paper, they used three concentrations of DON in each group during their trial. However, we only used one concentration of DON and the same feed in each group throughout the experiment. So the two papers acquired different results on these cytokines levels.

Point 2: The sections 2.3, 2.4 and 2.5, and the caption of Figure 4 repeat similar sentences and appear redundant. Could the authors improve these texts a little more?

Response 2: Thank you for your valuable suggestions. The sections 2.3, 2.4 and 2.5, and the caption of Figure 4 has been simplified.

This manuscript is a resubmission of an earlier submission. The following is a list of the peer review reports and author responses from that submission.

Round 1

Reviewer 1 Report

This is very important study describing DON induced intestinal damage through NF-κB signaling pathway in weaned piglets. The authors investigated the mechanism of DON in vivo and suggested that this mycotoxin may induce inflammation in small intestine tissue by damaging NF-κB signaling pathway. 

Despite the importance of the article, I would like to point out some general points to improve the article. 

Extensive editing of English language and style required

In the Introduction part - please expand on the importance of pro-inflammatory cytokines

The quality of the figures must be improved (currently all figures appear under low resolution). In figures 3 and 5 - y axis title - please change to "relative expression"

In the Results part - not only informative data should be presented, such as "The relative expression... was significantly lower", and in the next paragraph: "The relative expression... was significantly increased", and so on. Lack of the simple discussion and interpretation of the results (it's basically about improving the style).  

Reviewer 2 Report

General:

Bad English language – sometime hard to figure out the sense

Many spelling and grammar mistakes

I would suggest rejecting the manuscript, since there are many things to improve.

Title:

Never use abbreviations in the title.

The keywords are repetitions of words in the title – that is also not correct.

Statistics:

You do not describe which tests were used. Below the figures you write “from three independent experiments (n=5)”. At the section Statistics it is writen “All data represent mean +/- SD (n=10)” à so what is correct?

Significance p = written in small letters

Analysis of staining was significant? How did you calculate that?

Results:

At figure 2, you mention a negative control – what is that? It is not explained somewhere.

Figure 3: wrong description of figure – you mention D and E and F – but there are no graphs

Same is true for Figure 5 – maybe you mixed that up?

Material:

Did you analyze the diet for mycotoxins before including DON into the diet? And how did you contaminate your diet with the two DON concentrations?

Discussion:

There is actually no discussion with your own results. It is more a repetition of the introduction. It is too short for the amount of data. Too less references used.

References:

Only 28 references, the important references are missing. For example: in the introduction you write “DON has various toxic effects such as neurotoxicity, cytotoxicity, etc.” and then you refer to only 1 paper.

Reviewer 3 Report

Dear Authors,

The topics of the presence of mycotoxins in feed and food is a very important in animal production and public health. Although the mechanism of action of a large group of these compounds has been described, animal studies are a very important element of scientific research, and can be a direct confirmation of the mechanism in vivo action.

The evaluated manuscript "DON induced intestinal damage through NF-κB signaling pathway in weaned piglets" refers to the effect of the presence of high concentrations of deoxynivalenol on selected parameters of gastrointestinal inflammation of pigs and intestinal integrity. However, the manuscript contains numerous errors and shortcomings. It is also a duplication of previous research, only in the original combination of parameters and doses used. There is no explanation of the deoxynivalenol concentrations used, no description of way of the toxin use, no description of feed intake. In toxicological terms, the concentration is given in μg / kg b.w. and not in mg / kg of feed (it's best to give both parameters).

In the manuscript it is not described what was the feed intake. The experiment was carried out for a relatively long time, with the changing age of animals-the adoption of fodder is also growing. There is also no information about the test of mycotoxin content in the feed used (a high content of maize). The material also contains figures with  poor quality ( it is difficult to assess them in any way). There are also numerous errors in the signature of the figures. In its present form, manuscript does not present any scientific value, therefore I suggest its rejection.

Detailed comments:

Line 2 - the Title does not correspond to the contents of the manuscript

Line 6 - the mechanism of action DON is well described –e.g. „From the gut to the brain: journey and pathophysiological effects of the food-associated trichothecene mycotoxin deoxynivalenol. Toxins  2013 Apr 23;5(4):784-820“.

Line 12 - "OD values" - explain the abbreviation used for the first time

Line 18 - "weaned piglets" - the experience was 60 days long, so on the day of slaughter they were not a weaned piglets.

Line 25 - graphic abstract is chaotic

Line 30 - should be fusarium spp. and other types

Line 32 - not only China - worldwide.... why the finished feed were not analyzed?

Line 36 "limimted toxicity".? what Authors means?

Line 37 - this is not „beliwe”- it is knowledge ....

Line 40 - IEC 6 is a rat cell line!

Line 45 - no sense in the sentence

Line 46 - transcellular?

Lina 54 - progresing?

Line 59 - no sense in the sentence

Line 68 - no Figures A1 in the manuscript

Line 74 - were observed?

Line 90 - unreadable figure

Line 102 - Fig. 3B – this same error in many places.

Line 114 - signature incompatible with the contents of the drawing!

Line 187 - signature incompatible with the contents of the drawing!

Line 195- Intstinal mucosa is build by intesstinal ephitelial cells.

Line 197 - "lymphocytes in intestinal villus epithelial cells" - erroneous statement

Line 203 - "usually"? and in what cases not?

Line 254 - the use of DON in such quantities purchased in SIGMA Chemical Co - it raises serious doubts.

Line 302 - indicated application of two house keeping genes.

Line 325 - no counting / analysis method.